# Late Neurological Consequences of Zika Virus Infection: Risk Factors and Pharmaceutical Approaches

**DOI:** 10.3390/ph12020060

**Published:** 2019-04-17

**Authors:** Isis N. O. Souza, Fernanda G. Q. Barros-Aragão, Paula S. Frost, Claudia P. Figueiredo, Julia R. Clarke

**Affiliations:** 1School of Pharmacy, Federal University of Rio de Janeiro, Rio de Janeiro 21944-590, Brazil; isis.de.o@gmail.com (I.N.O.S.); fernandaqbarros@gmail.com (F.G.Q.B.-A.); paulafrostufrj@gmail.com (P.S.F.); 2Institute of Biomedical Sciences, Federal University of Rio de Janeiro, Rio de Janeiro 21944-590, Brazil

**Keywords:** microcephaly, arbovirus, flavivirus, drug development, neurodevelopment, therapy, drug screening, pregnancy, Food and Drug Administration (FDA), encephalitis

## Abstract

Zika virus (ZIKV) infection was historically considered a disease with mild symptoms and no major consequences to human health. However, several long-term, late onset, and chronic neurological complications, both in congenitally-exposed babies and in adult patients, have been reported after ZIKV infection, especially after the 2015 epidemics in the American continent. The development or severity of these conditions cannot be fully predicted, but it is possible that genetic, epigenetic, and environmental factors may contribute to determine ZIKV infection outcomes. This reinforces the importance that individuals exposed to ZIKV are submitted to long-term clinical surveillance and highlights the urgent need for the development of therapeutic approaches to reduce or eliminate the neurological burden of infection. Here, we review the epidemiology of ZIKV-associated neurological complications and the role of factors that may influence disease outcome. Moreover, we discuss experimental and clinical evidence of drugs that have shown promising results in vitro or in vitro against viral replication and and/or ZIKV-induced neurotoxicity.

## 1. Introduction

Zika virus (ZIKV) has been known since 1947 [1], and was historically regarded as a geographically restricted virus causing minor symptoms in humans. In 2013, this view started to change when ZIKV occurrence in French Polynesia was associated to development of Guillain-Barre syndrome, a serious autoimmune condition affecting the peripheral nervous system. Only two years later, the virus landed in the Americas, where it was responsible for a major outbreak becoming a serious public health concern. It is predicted that over 800,000 people were infected by ZIKV in the American continent between 2015 and 2018 [2]. Following this epidemic, several studies endorsed a causal relationship between ZIKV infection and neurological disorders, both in congenitally-exposed newborns and in adult patients. The development or severity of these conditions cannot be predicted based on genetic or environmental factors, which reinforces the importance of long-term clinical surveillance of individuals exposed to ZIKV. Here, we review the epidemiology of ZIKV-associated complications and discuss experimental and clinical evidence of potential therapeutic approaches.

## 2. ZIKV Infection Is Associated to Late Detrimental Effects to the Developing and Mature Nervous Systems

A spectrum of neuropathological conditions has been reported in newborn babies exposed to ZIKV in uterus, being microcephaly the most severe outcome. Infants who develop ZIKV-induced microcephaly show recurrent seizures [3], and severely compromised neuropsychomotor development is frequently reported [4]. Several alterations were described after brain tomography of babies congenitally exposed to ZIKV, including widespread calcifications, cerebral atrophy, brainstem and cerebellar hypoplasia, as well as ventriculomegaly [5] (Figure 1A). 

Extensive research has focused on unraveling the mechanisms of congenital microcephaly induced by ZIKV [6,7,8]. However, increasing evidence from clinical and experimental studies suggest that even infants considered healthy at birth can develop severe long-term neurological complications as a consequence of exposure. According to US hospital records, 6% of children infected with ZIKV in uterus showed malformations detectable at birth. Importantly, the percentage of infants with ZIKV-associated neurological complications rose to 14% when babies considered normal at birth were followed-up after leaving hospital [9]. Among the most common symptoms reported were personal-social developmental delays, swallowing difficulties, and a myriad of motor dysfunctions including compromised mobility, muscular hypertonia, dystonic movement, and impaired fine motor skills [9,10,11]. Post-natal onset microcephaly was also reported, and studies suggest that 60% of normocephalic babies exposed to ZIKV in uterus present seizures at some stage of development [11]. Several alterations were also described in brains of normocephalic ZIKV-exposed babies assessed by neuroimaging tools, including subcortical calcifications, increased frontal cerebrospinal fluid space and ventricle enlargement [12]. Infants exposed to ZIKV during the epidemic in the American continent are still under their third year of life, making it impossible to predict which of these symptoms may persist and if others may emerge. Animal models are useful tools to investigate the mechanisms of infection and its complications, and whereas most studies have focused on unraveling the mechanisms of ZIKV-induced microcephaly, a few others have provided insight into the mechanisms underlying late ZIKV-induced neurological damage. Using a mouse model of ZIKV infection, our group was able to predict possible long-term alterations in children after a congenital infection. Since the stage of development of the mouse brain at early postnatal period resembles the events taking place in the human brain during the second and third trimester of pregnancy [13,14,15], we performed ZIKV infection shortly after birth (post-natal day 3). Indeed, we found that infected pups presented neuropathological hallmarks that closely resembled those seen in brains of babies after ZIKV congenital infection. Over 90% of mice showed spontaneous seizures during childhood, and intensity of seizures decreased as animals grew into adulthood. Moreover, even though these episodes resolved when animals became adults, mice infected neonatally with ZIKV remained more susceptible to chemically-induced seizures. Besides, ZIKV-infected mice showed motor and cognitive disabilities in adulthood. Similar results were obtained in another study, in which authors demonstrated that early-life infection with an African ZIKV strain results in persistent memory impairments in mice [16]. These findings were further confirmed in non-human primates infected in uterus or shortly after birth. As adults, these neonatally-infected macaques developed an atypical emotional response [17] and neuropathological characterization of brains showed reduced white matter volume, ventriculomegaly, and decreased hippocampal growth in ZIKV-infected animals. 

Neurological complications following ZIKV infection are not limited to cases of congenital infection [18]. Mounting clinical evidence show that ZIKV infection in adult patients may result in encephalitis [18,19,20], encephalomyelitis [18,20], acute myelitis [18,19], chronic inflammatory demyelinating polyneuropathy [21], and Guillain-Barre syndrome [19,22,23] (Figure 1B). Episodes of septic shock were also reported, and in rare cases these complications were shown to be fatal [21,24]. In addition, ZIKV-associated Guillain–Barre syndrome may, in some cases, lead to development of chronic pain [18]. One study found that 40% of patients still showed some kind of neurological alteration after 1 year of symptom onset, and after two years some patients had not yet fully recovered [25]. Collectively, these reports demonstrate that ZIKV infection is also detrimental to the mature nervous system and is associated with a spectrum of neurological syndromes that may culminate in long-term consequences and even mortality. Whether these neurological consequences are a direct effect of ZIKV replication or are secondary to the host immune response still need to be investigated; a question that could lead to important clues on how to prevent long-term consequences of infection. Different research groups have shown beneficial effects of both antiviral and anti-inflammatory strategies in preventing long-term behavioral consequences of infection. These findings are reviewed in the following sections.

Another matter of concern involves the ability of ZIKV to persist in certain organs and tissues, since the late consequences of viral persistence remain unknown. ZIKV usually reaches detectable levels in the serum of patients within 3–10 days after onset of symptoms, shortly returning to undetectable levels [26]. Even after viral clearance from blood and up to 200 days after infection, detectable levels of ZIKV RNA were found in placenta of pregnant women [27]. ZIKV was also shown to persists in semen samples up to 93 days after the onset of symptoms, reinforcing the high risk of sexual transmission [28,29]. ZIKV RNA was found in brains of babies exposed in utero on average 163 days after infection, and one study described high viral loads after post-mortem analysis of the brain of a 5-month-old infected infant [5]. The ability of ZIKV to replicate for long periods in specific tissues has also been demonstrated in animal models. In adult non-human primates infected subcutaneously with ZIKV, virus remained in neuronal tissue, cerebrospinal fluid, and lymph-nodes, while in the serum levels were undetectable after only seven days post-infection [30,31]. Further evidence of persistent replication in neuronal tissue were found in immunocompetent mice infected during the neonatal period, where ZIKV negative RNA strand was detectable in the brain into adulthood (100 dpi), an indicative of active viral replication [32]. Altogether, these results provide evidence that ZIKV is capable of persisting in certain tissues long after resolution of symptoms and clearance of virus from blood. Factors that may contribute to viral persistence remain unknown, as well as if new rounds of active viral replication and reoccurrence of symptoms can occur at later stages of life, including moments of compromised immune response or as consequence of other viral infections.

## 3. What Genetic, Epigenetic, and Environmental Factors May Increase the Risk of ZIKV-Induced Neurological Damage?

Little is known about what factors can contribute to the development of late neurological damage associated to ZIKV infection, and what are the precise mechanisms involved. It is likely that several genetic and environmental factors interplay to increase one’s susceptibility to the development of neurological outcomes (Figure 1). A comprehensive knowledge of how ZIKV-induced neurological damage is induced in adult and infant patients is crucial to prevention and treatment of such severe sequelae.

ZIKV strains isolated from different geographical regions are derived from two main lineages: African and Southeast Asian [26]. Even though the African strain is more cytotoxic and neurovirulent in experimental settings [33,34], the ZIKV strain responsible for the outbreak in the Americas is genetically closer to strains isolated in French Polynesia in 2013 and belongs to the Southeast Asian subclade [35]. One study has shown that the contemporary strain (isolated in 2015–2016) carries several nucleotide substitutions when compared to a strain isolated in 2010 in Cambodia. A serine-to-asparagine substitution (S139N) in the Pr domain of the viral precursor membrane protein (prM), was shown to be especially responsible for its increased neurovirulence [36]. Interestingly, this substitution emerged in the Southeast Asian ZIKV strain shortly before the 2013 outbreak and was maintained in the subsequent epidemics in the Americas [36]. 

Mutations in ZIKV genome may also promote host immune system evasion and enhance infectivity of mosquito hosts [37,38], which could contribute to increase the geographical spread of the virus. African and Asian populations were highly exposed to ZIKV historically [39,40,41,42,43], and it has been speculated that prior contact to the virus may provide protective immunity [44]. Thus, it is reasonable to speculate that the sudden exposure of an immunologically naïve population, in the American continent, provided unseen absolute numbers of acutely ZIKV-infected patients, favoring the observation of long-term consequences.

Variations on the hosts’ genetic background are determinant to an appropriate immune response, and may thus influence the occurrence of ZIKV-associated neurological complications. Experiments in animal models support this notion, since infection is associated to either mild or severe signs of disease, depending on the genetic characteristics of the host. For example, effective systemic infection in mice requires a limited interferon-mediated immune response, which is obtained either by knocking out specific components of interferon (IFN) pathway [45,46,47] or by using neonatal mice [32], a stage where interferon-mediated immune response is lacking or insufficient. While type III IFN response is protective against ZIKV infection [48], type I response is related to miscarriage and growth restriction, especially during mid-gestation [49]. Several signaling pathways were shown to modulate ZIKV infection and individual differences in their expression may confer variable risk to congenital Zika syndrome (CZS) and other neurological outcomes: TAM receptors (TYRO3, AXL, and MER) are important in mediating ZIKV persistence through type I IFN downregulation [50,51,52,53,54,55]; toll-like receptors 3 (TLR3) increase viral replication and are important mediators of inflammatory responses [50]; infection triggers autophagy pathways, but their effects on viral replication and immune response are still controversial [50,56] (for review see [57]). In addition, the chances of prior or concomitant infection by other mosquito-borne viruses is high in ZIKV endemic regions and co-infections might modulate the hosts’ immune response either in a protective [58,59] or detrimental way [60,61,62,63,64]. Experimental evidence suggest that mosquitoes previously infected with Chikungunya virus have enhanced ZIKV infectivity and vector competence [65].

In line with this, individual responses determined by either genetic or epigenetic backgrounds have been shown to modulate susceptibility of cells in the developing brain to ZIKV. In a recent study, neural stem cells (NSCs) derived from three independent donors were infected in vitro with ZIKV (Mexico isolate). The virus was shown to replicate and induce decreased cell proliferation in all NSCs strains. However, NSCs from one of these donors were resistant to ZIKV-induced decrease in neuronal differentiation. RNA expression patterns changed accordingly: ZIKV-sensitive strains showed pronounced reduction in pathways related to neurogenesis and higher expression in pathways responsible for innate immune response [66]. In another interesting study, neural progenitor cells (NPC) were induced from salivary cells obtained from twins exposed to ZIKV in uterus [67]. Even though the study was performed in a small cohort, both concordant (both fetuses showing CZS) and discordant (only one fetus with CZS) twins were analyzed. No unique genetic trait that could explain differential susceptibility of siblings to developing CZS was found, therefore, authors suggest there could be multifactorial inheritance or even epigenetic modifications linked to different responses to viral infection. In line with this, NPCs and neurospheres derived from CZS-fetuses produced more infectious particles when exposed to ZIKV in vitro than those from non-CZS infants. RNA sequencing performed in these cells reported changes in expression patterns, notably upregulation of mammalian target of rapamycin (mTOR) and Wnt pathways, exclusively in affected twins [67].

Concerning the development of CZS, it can also be speculated that a pregnant woman’s individual response to infection influences the efficiency with which ZIKV crosses the placental barrier. Trophoblasts, placental barrier cells, are permissive to ZIKV only during a narrow time-frame, likely in the first trimester of gestation [48,53,68,69,70]. This is probably one of the reasons why infection during this stage of pregnancy is usually associated to the birth of babies with more severe neurological outcomes. Trophoblasts secrete type III IFN [48,70], and the amount of type III IFN released could be related to the efficacy of ZIKV infection in these cells. In addition, other events known to limit IFN response, like concomitant viral infections, could enhance ZIKV infectivity. Accordingly, concomitant infection with Herpes Simplex 2 virus (HSV-2) increases susceptibility of trophoblasts to ZIKV [53]. Alternatively, to direct transport, it is possible that maternal antibody-mediated transcytosis drives ZIKV transportation across the placental barrier. Indeed, experimental evidence has shown that placental ZIKV replication is enhanced by previous infection with Dengue virus [60,71]. In addition to trophoblasts, placenta villous macrophages (Hofbauer cells) and fibroblasts are susceptible to ZIKV infection [68,72]. When exposed to ZIKV, Hofbauer cells become activated, but the extent of activation and cytokine production is highly variable between individuals. It is thus possible that the intensity of placental immune response influences how much ZIKV actually reach the fetus [68]. 

Despite factors specific to host and viral genomes, several other factors can have contributed to the unforeseen scale of ZIKV neurological damage seen in recent years. An interesting study in immunodeficient mice has suggested that sexual transmission of ZIKV favors congenital infection and the development of malformations of fetuses, suggesting that the route of infection may also play a role in the severity of congenital ZIKV syndrome [73].

As recently reviewed by Barbeito-Andrés and colleagues [74], environmental stresses emerging from adverse social-economic conditions or other geographical factors can contribute to neurological outcome of infection. In Brazil, although the incidence of ZIKV infection was similar in all regions in 2015, congenital ZIKV syndrome was significantly higher in the northeast region, a severely impoverished region [75]. When comparing different regions of the northeast of Brazil, epidemiological studies showed that higher incidence of microcephaly was observed in districts with lower developmental index, higher rates of mosquito larvae density, and more inefficient sewage and garbage collection systems [76]. Impoverished populations are more frequently susceptible to mal-nutrition or inappropriate diet, degrading housing conditions which facilitates mosquito spreading and disease transmission, higher incidence of coinfections and poorer access to health system and family planning, among others. Additional population-based and experimental studies should be performed in order to determine whether and how any of these factors has contributed to determine the higher rate of neurological complications due to ZIKV infection in specific geographical regions.

## 4. Promising Therapeutic Approaches Targeting ZIKV-Associated Symptoms

Similar to other mosquito-borne diseases, clinical management of patients with ZIKV infection is largely based on alleviating symptoms of the acute phase of infection. The American Center for Disease Control and Prevention (CDC) and the World Health Organization recommend generic measures such as resting, drinking fluids, and using over-the-counter medication to alleviate pain and fever [77,78]. However, considering the serious long-term damages that have been associated to ZIKV, it is increasingly clear that these patients need closer follow-up. Experimental and clinical studies have focused on testing prophylactic drugs and creating vaccines to prevent new ZIKV epidemics. Nevertheless, some effort has also been directed to preventing development of ZIKV-associated late complications, especially due to intra-uterine exposure of the virus. In this sense, drug repurposing is an especially promising approach, since it surpasses several steps of pre-clinical drug development [79]. In this section, we review evidence of several antiviral and disease-modifying drugs, either repurposed or under development, which have shown promising results in different models of ZIKV infection. These substances act either during acute stages of infection by inhibiting cell invasion or viral replication, or have shown potential to prevent the late complications of ZIKV infection (Table 1 and Appendix A). 

## 5. Antiviral Drugs 

### 5.1. Sofosbuvir

Early reports using in silico and in vitro approaches suggested the potential of sofosbuvir (SFB), an antiviral prodrug clinically used against hepatitis C virus (HCV), to inhibit ZIKV replication [100,101]. Later, two independent groups published experimental evidence of anti-ZIKV activity of this drug. In these studies, it was shown that SFB effectively inhibited infection and replication by both African and American ZIKV strains in different in vitro models: placental and neuroblastoma cell lines, human fetal-derived neuronal stem cells, induced-pluripotent neural stem cells, and brain organoids [102,103]. It is a consensus that SFB’s mechanism of action consists on blocking the highly conserved RNA polymerase of flaviviruses [103]. However, in vitro findings show that SFB increases the frequency of viral genetic mutations, indicating a possible secondary mechanism of ZIKV replication blockage. Experiments performed in animal models also showed promising results, since administration of SFB in drinking water (33 mg/kg/day) to 5 weeks-old C57BL/6 mice infected with the African viral strain prevented ZIKV-induced mortality and weight loss [102]. Similar results were obtained with neonatal Swiss mice infected at post-natal day 3 and treated intraperitonially for seven days with 20 mg/kg of SFB, starting either 1 day before or 1 day after infection [16]. Furthermore, SFB was capable of decreasing viral load in brain and peripheral structures and partially rescued late ZIKV-induced memory impairment in two different behavioral paradigms [16]. When SFB was used in association with interferon-α in a human liver cell line, a synergic antiviral activity was observed against several different strains of ZIKV [105], proving that a combination strategy might be beneficial. SFB is also an FDA category B drug, being considered generally safe for use during pregnancy based on preclinical studies. A Phase 1 clinical trial is currently being performed in order to evaluate the safety of SFB administration during pregnancy, and is scheduled to end by 2020 [106]. Although mathematical modeling indicated that the expected dose needed for ZIKV inhibition would be twice the therapeutic dose used for HCV infections, it would still be below its toxicity threshold [107,108]. However, its high price might be a limitation to allow its widespread use in low- and middle-income countries.

### 5.2. Favipiravir

Another RNA polymerase inhibitor that has been proposed as ZIKV therapeutic is favipiravir (FAV; T-705). Early reports indicated its ability to inhibit seven different ZIKV strains in preliminary tests in vitro [88,89]. Conflicting results, however, were found in studies designed to evaluate whether FAV inhibits ZIKV replication in more complex models. While one group reported no effect on viral replication in either cortical neurons, motor neurons or astrocytes derived from induced-pluripotent stem cells (iPSCs) [90], another study showed that FAV induced efficient reduction in ZIKV protein expression in human neural progenitor cells derived from human embryonic stem cells (hESCs) [91]. This is likely due to the different source and/or differentiation stage of the stem cells used and the specific protocols used for each derivation. FAV has been shown to cause significant mutation in ZIKV genome, indicating yet another possible mechanism for its antiviral activity [109]. When combined with interferon, a synergic effect arises, guaranteeing extended viral inhibition without signs of cytotoxicity in Vero cells [110]. In silico models of ZIKV infection using non-human primate pharmacokinetic parameters, indicate that 150 mg/kg of FAV given twice daily is enough to reduce viral load, efficiently shortening disease duration [111]. Given its good tolerability in clinical trials [112], FAV is a strong candidate for further pre-clinical and clinical testing for ZIKV treatment. Furthermore, its analog, T-1105, has also presented antiviral activity against SZ01 ZIKV strain [89], indicating it might be a good scaffold for drug development.

### 5.3. Azithromycin

Retallack and colleagues used a drug repurposing strategy to identify the macrolide antibiotic azithromycin (AZT) as a potential anti-ZIKV drug. AZT was found to be the most promising drug, out of 2177 clinically-approved compounds, in preventing infection of the glial cell lineage U87 by rescuing cell viability [82]. In another study, the efficacy of the compound was tested in a mouse model of neonatal infection. AZT (10 mg/kg i.p.)-treated mice showed decreased mortality, better clinical score, fewer viral RNA copies in the brain and reduced monocyte infiltration following ZIKV infection [83]. AZT is an FDA category B drug, which means it is considered generally safe for use during pregnancy based on animal studies [113]. Moreover, AZT reaches high plasma levels and has a long half-life in human brain and placental tissue, indicating that it is a potential candidate to treat infections in both adult subjects and during the perinatal period [114,115]. Finally, its widespread availability in several formulations and considerably low cost makes AZT an interesting candidate for post-exposure ZIKV treatment. No studies so far have investigated the precise mechanism of action of this drug against ZIKV.

## 6. Disease-Modifying Drugs

### 6.1. Memantine

Excessive glutamate release from synaptic terminals causes an overactivation of the nearby neuronal network leading to neuronal damage and death. Such phenomenon, named excitotoxicity, is a common feature of several neurological diseases [116]. Considering the extensive cell death caused by ZIKV, it is reasonable to hypothesize that glutamate released from infected/dying neurons could cause excitotoxicity in the brain and contribute to further aggravate neurodegeneration [94]. Based on such assumption, Costa and colleagues showed the neuroprotective effect of several N-methyl D-Aspartate (NMDA) glutamate receptor blockers, including MK-801, agmatine sulfate, ifenprodil, and memantine (ME), in models of ZIKV-induced neuronal damage both in vitro and in vitro. Interestingly, all substances were able to prevent glial and neuronal death in primary cell cultures, even though viral load were unchanged. ME’s beneficial effect was further evaluated in mice lacking interferon-mediated immune response (IFN-α/βR^−/−^ mice) infected with the Brazilian ZIKV strain: daily oral treatment with 30 mg/kg of ME from days 3 to 6 post-infection prevented ZIKV-induced increase in intraocular pressure, ameliorated brain histopathological lesions and led to decreased microgliosis [94]. ME is an FDA category B drug approved to treat mild to severe Alzheimer’s disease. It is available in several oral formulations with good safety profile [117]. Further studies are necessary to determine its efficacy in ZIKV management in other animal models and in human subjects.

### 6.2. Infliximab

Considering the extensive pro-inflammatory profile of brain tissue during and after ZIKV infection, anti-inflammatory therapy might be of help to alleviate further detrimental consequences. Our group has recently evaluated whether the TNF-α neutralizing antibody infliximab (IFX) (20 μg/day) is able to prevent long-term consequences of ZIKV infection in neonatal (post-natal day 3) Swiss mice [32]. IFX treatment induced a 50% decrease in spontaneous seizures induced by infection and rescued sensitivity to seizures induced by chemicals in adulthood. However, ZIKV-induced memory and motor impairments in adult mice were not prevented by the TNF-α blocking antibody. An important limitation to therapeutic use of IFX is the increased risk to develop malignancies, such as lymphoma, when used for long periods in pediatric patients [118]. Then, despite the security of IFX therapy in adult patients, extensive safety studies and careful cost-benefit analysis must be performed before its use is indicated for ZIKV-exposed children

### 6.3. PHA-690509

Xu and colleagues were one of the first groups to make a large drug repurposing screen for ZIKV, obtaining exciting results [87]. One of the most interesting substances identified in this study was PHA-690509 (PHA), a cyclin-dependent kinase inhibitor, a drug that blocks the activity of several kinases involved in cell cycle regulation that is currently considered a potential drug candidate to treat ZIKV infection. PHA showed antiviral activity against two different ZIKV strains in human astrocytes derived from iPSCs and human neural progenitor cells (hNPCs). Twenty-seven other structurally related cyclin-dependent kinase inhibitors were also tested, nine of which presented submicromolar IC_50_ concentrations against ZIKV replication in culture. These findings open new avenues for drug design and in silico optimization. However, because viruses do not present cyclin-dependent kinases, it is possible that the effect of such drugs rely on the regulation of the host cell differentiation machinery. Therefore, such compounds might not be indicated for pregnant women as fetus development might be affected [119].

### 6.4. Emricasan

Emricasan is a pan-caspase inhibitor which was shown to increase viability of human astrocytes and hNPCs infected with several different ZIKV strains [87]. This strong neuroprotective effect was not due to inhibition of ZIKV, suggesting that its mechanism of action is independent of viral replication inhibition. Interestingly, an additive effect was found when emricasan was used in combination with the cyclin-dependent kinase inhibitor PHA, significantly improving cell viability of human astrocyte cultures exposed to ZIKV when compared to the effects of both isolated substances. Emricasan was well tolerated in phase 1 studies and recently entered phase 2 clinical trial against hepatic consequences of HCV infection [120], being a good candidate for ZIKV management, especially in combination with other drugs.

### 6.5. Niclosamide

The screening performed by Xu and colleagues also identified the anti-helmintic drug niclosamide as a powerful antiviral drug against ZIKV, being able to mitigate infection in a human glioblastoma cell line (SNB-19) and in human astrocytes (Xu et al., 2016). In another study, two doses of niclosamide (on embryonic days 3 and 5; 50 mg/kg) were tested in a chick embryo model of ZIKV-induced microcephaly. Niclosamide was able to partially rescue ZIKV-induced reductions in cranial size and significantly improved brain inflammatory profile [95]. Niclosamide is an FDA class B drug that has been used for over 50 years with good safety profile [121]. Its mechanism of action against other viruses consists of endolysosomal pH neutralization, inhibiting viral entry in cells [122]. However, time-of-addition experiments indicate that niclosamide is still effective even after viral cell invasion is finished, suggesting that there might be other mechanisms by which the drug mitigates ZIKV infection. Several groups have developed innovative formulations in order to bypass its poor oral bioavailability [123,124,125]. It also has good safety profile, being considered an FDA category B drug [126]. Toxicology assessment in zebrafish indicate potential teratogenic effects of niclosamide, which must be closely evaluated before clinical studies [127]. Therefore, niclosamide seems like a good candidate for management of infection in adult non-pregnant patients and its use in pregnant women must be weighed. 

### 6.6. Chloroquine

Chloroquine (CQ) is a classically used antimalarial drug. Its antiviral activity against dengue virus and other flaviviruses has already been demonstrated [128,129] and several groups started to test its possible anti-ZIKV activity. Delvecchio and colleagues tested CQ against African and Brazilian strains of ZIKV in human neural progenitor cells, human brain microvascular endothelial cells and mouse neurospheres, indicating its ability to inhibit viral replication and rescue cell viability (Delvecchio 2016). CQ also reduced mortality and weight loss caused by ZIKV infection in adult AG129 mice, when administered in drinking water at the dose of 50 mg/kg for two days before infection and five days after, with a maintenance dose of 5 mg/kg until the end of the experiment [85]. In that same study, authors showed that vertical transmission was attenuated when pregnant dams received 30 mg/kg/day of CQ in their drinking water, resulting in decreased levels of viral RNA and ZIKV-positive cells in the brains of fetuses. These results were further replicated in another study, which employed a similar treatment schedule: when given to pregnant BALB/c dams (20 mg/kg/day s.c.) between embryonic days 13.5 and 18.5, CQ was able to reestablish cortical thickness, reduce caspase-3 immunolabeling and rescue the expression of several proliferation markers in ZIKV-infected embryos [86]. CQs safety profile during pregnancy is outstanding in every trimester as well as with prolonged use. Human studies performed in lactating women reported no abnormalities [130]. Moreover, CQ is extensively available worldwide and its low cost makes it a good candidate for both prophylaxis and treatment of ZIKV-infected men and women.

### 6.7. Amodiaquine

Another interesting antimalarial quinine with anti-ZIKV activity is amodiaquine (AQ). A high-content screening study identified its ability to prevent ZIKV infection and its toxicity in human pluripotent stem cell-derived neural progenitors [80]. Moreover, it showed comparable results to CQ when tested against Asian PLCal_ZV ZIKV strain in Vero cells [81]. In an adult immunodeficient mouse model, seven days of treatment with AQ (40 mg/kg/day s.c.) led to lower ZIKV RNA levels and decreased levels of ZIKV E-protein in the brain [80]. Three AQ derivatives, quinacrine, mefloquine, and GSK369796, were also able to inhibit viral replication with an EC_50_ within the micromolar range, indicating that AQ is also an interesting scaffold molecule for drug development (Balasubramanian, 2017). AQ has been used for treatment of malaria in pediatric settings, but serious adverse effects were shown after its long-term use for prophylaxis [130]. Therefore, studies should focus on its development as short-term prophylactic drug or post-exposure treatment.

### 6.8. NITD008

Adenosine analogs share the same mechanism of action which consists of impairing the function of viral RNA polymerase of several viruses such as measles and rabies [131]. One specific adenosine analog, NITD008, has been proven effective against a number of flaviviruses, raising the hypothesis that it would be a promising anti-ZIKV drug [98,132]. NITD008 was able to decrease viral plaque forming units and diminish genome replication in culture lysates when added to Vero cells infected separately with two different strains of ZIKV [96,97]. Previous reports had already indicated that NITD008 has good oral bioavailability and pharmacokinetic profile as well as low in vitro and only short-term in vitro toxicity [98]. Treatment of adult ZIKV-infected A129 mice with NITD008 (5 doses of 50 mg/kg i.p.) decreased mortality and blood viremia. Further investigation of its safety profile is needed before it is considered as an anti-ZIKV drug.

### 6.9. Galidesivir (BCX4430)

Another drug with similar mechanism of action is BCX4430 (Galidesivir), which showed strong antiviral activity against yellow fever and several others viruses in previous studies, and was recently tested against ZIKV [133]. It was able to reduce cytopathic effect and viral load in Vero76, Huh7 and RD cell lines infected with African, Asian, or American ZIKV strains [92]. Furthermore, in a model of lethal infection in adult immunodeficient mice, treatment with BCX4430 for 8 days, starting 4 h prior to infection was able to improve survival and prevent infection-induced weight loss. Similar results were obtained when the drug was given 1 or 3 days post-infection [92]. A preliminary study in Rhesus monkeys indicated good tolerability and efficacy of the candidate when given intramuscularly at a dose of 100 mg/kg on the day of infection followed by 25 mg/kg twice daily for 9 additional days [93]. As of January 2019, BioCryst Pharmaceuticals was recruiting for a Phase 1 human trial in order to test single dose safety, tolerability and pharmacokinetics of BCX4430 [134].

### 6.10. Ribavirin

Ribavirin (RIB) is a nucleoside analogue used for the treatment of hepatitis C. In an early report, RIB was able to inhibit replication of several different ZIKV strains in vitro [88]. Such result was confirmed by independent groups and a mechanism of mutagenesis was also proposed [99,109,135]. Conflicting results were obtained when RIB was tested in stem cells: while no inhibition of viral replication was found in iPSC-derived cells [90], promising results were found in hNPCs [91]. Nevertheless, in immunocompromised mice where infection leads to 100% mortality, administration of high doses (15 mg/day) of RIB for 3 consecutive days post-infection was able to partially suppress viremia and prolong survival for a few days [99]. RIB should not be used by pregnant women or by their male partners due to right risk of fetal death and birth defects. Also, it presents several adverse effects when administered in high doses or in a prolonged setting [136]. Therefore, its use might be limited to severe consequences of ZIKV infection exclusively in otherwise healthy adults.

### 6.11. Other Reported Candidates

Z2 is a synthetic peptide derived from the stem region of ZIKV envelope protein which showed potent anti-ZIKV activity. Z2 was able to inhibit infection and diminish cytotoxic effects of ZIKV on Vero and BHK21 cells. In pregnant ICR mice, Z2 was able to penetrate the placental barrier and cause no visible birth defects or perinatal damage in doses up to 120 mg/kg. It also blocked vertical transmission of ZIKV in this same mouse model at 10 mg/kg i.v. In both A129 and AG6 mouse models, Z2-treated mice showed decreased ZIKV-induced mortality and smaller viremia [104]. Given the promising results, Z2 should be considered for preclinical and clinical tests. On the other hand, bithionol, an antihelmintic drug, and hippeastrine, an experimental alkaloid [80,137] have known cytotoxic profile in therapeutic doses, even though they showed good anti-ZIKV activity in vitro. Therefore, despite promising results, these drugs are poor candidates for future development. Finally, a single study proposed natural products as possible ZIKV inhibitors. The flavonoids Myricetin and Quercetin were able to inhibit the ZIKV NS2B-NS3 pro protein complex with sub- and micromolar concentrations [138].

## 7. Conclusions

The unprecedented high number of neurological symptoms associated to ZIKV infection after its landing in the Americas in 2015 has left severe economic and social consequences that cannot be fully estimated. The observation that ZIKV infection leads to late neurological burden both in congenitally-exposed infants and in adult patients has stimulated the international scientific community enormously. Researchers of different fields worldwide have gathered efforts in order to identify the mechanisms underlying ZIKV neurotoxicity and to search for drug candidates that can halt infection and neurological complications. In this sense, drug repurposing becomes an attractive alternative because several steps of drug approval by regulatory agencies can be surpassed. Drugs of different classes and mechanisms of action, including RNA polymerase inhibitors and several antimalarial quinines have been tested against ZIKV replication and its neurological damage. However, despite the promising in vitro and in vitro results, none of them has reached clinical trials for ZIKV infection yet. In silico drug design should also be employed to generate analogs with improved efficacy and pharmacokinetics.

In addition, further efforts should focus on investigating whether genetic and/or environmental factors contribute to neurological damage associated to ZIKV. Finding answers to this question will undoubtedly help defining public health strategies to prevent the occurrence of new epidemics with such devastating effects. In conclusion, the little available knowledge of factors that might contribute to ZIKV-associated neurologic complications and the lack of effective therapies approved to treat the acute and late stages of infection reinforce the importance of long-term clinical surveillance and follow up of individuals exposed to ZIKV.

## Figures and Tables

**Figure 1 pharmaceuticals-12-00060-f001:**
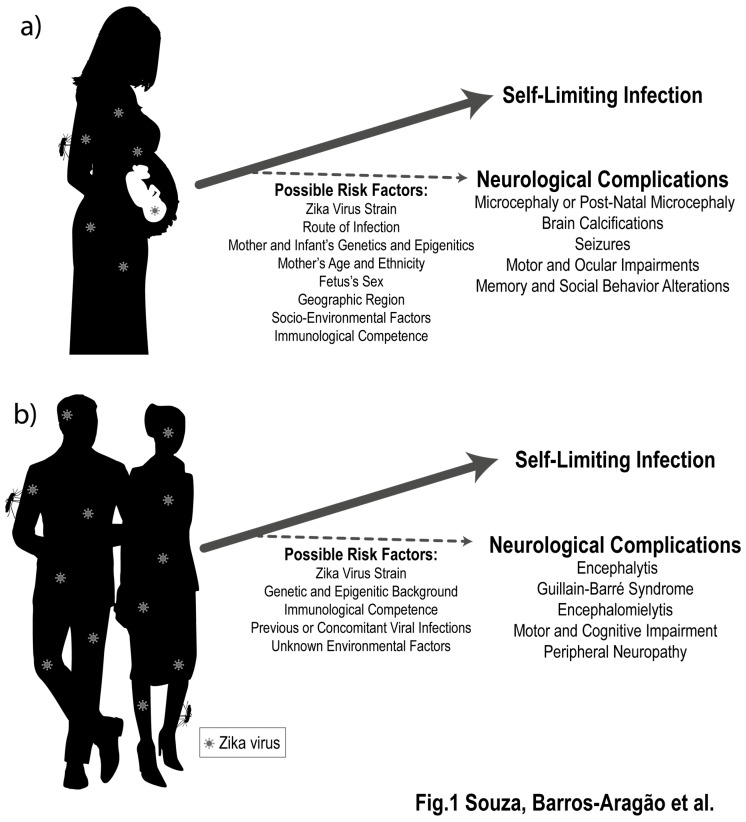
Zika virus (ZIKV) infection is associated to late detrimental effects to the developing and mature nervous systems. ZIKV infection during the perinatal period (**a**) and in adult patients (**b**) may be associated to late neurological complications. Although studies have raised possible contribution of several factors, the development or severity of these conditions cannot be fully predicted based on genetic or environmental factors, which reinforces the importance of long-term clinical surveillance of individuals exposed to ZIKV.

**Table 1 pharmaceuticals-12-00060-t001:** Drug Candidates for management of ZIKV infection and its late neurologic complications.

Drug	Proposed Mechanism of Action	In Vitro Efficacy	In Vitro Efficacy	Food and Drug Administration (FDA) Approved?	FDA Category	Refs.
Amodiaquine	Unknown for ZIKV (antimalarial)	Vero cell line human pluripotent stem cell-derived neural progenitors	40 mg/kg for 7 days s.c. to adult SCID-beige mice	Discontinued	-	[80,81]
Azithromycin	Unknown for ZIKV (antibiotic)	U87 glial cell line	10 mg/kg i.p. to neonatal ICR mice	Yes	B	[82,83]
Chloroquine	Unknown for ZIKV (antimalarial)	human neural progenitor cells human brain microvascular endothelial cells mouse neurospheres	50 mg/kg for 5 days + 5 mg/kg maintenance p.o. to adult AG129 mice 20 mg/kg for 6 days s.c. to pregnant BALB/c dams	Yes	Not assigned	[84,85,86]
Emricasan	pan-caspase inhibitor	human neural progenitor cells human astrocytes	N.D.	No	-	[87]
Favipiravir	RNA-dependent RNA polymerase inhibitor	Vero cell line induced-pluripotent neural stem cells human embryonic stem cells	N.D.	No	-	[88,89,90,91]
Galidesivir (BCX4430)	RNA-dependent RNA polymerase inhibitor	Vero76 cell line Huh7 cell line RD cell line	150 mg/kg for 8 days b.i.d. i.m.to adult AG129 mice 100 mg/kg + 25 mg/kg for 9 days b.i.d. i.m. to adult rhesus monkeys	No	-	[92,93]
Infliximab	TNF-α neutralizing antibody	N.D.	20 μg for 13 days i.p. to neonatal Swiss mice	Yes	B	[32]
Memantine	NMDA receptor blocker	glial primary cells neuronal primary cells	30 mg/kg for 4 days p.o. to adult IFN-α/βR^−/−^ mice	Yes	B	[94]
Niclosamide	endolysosomal pH neutralizer	human glioblastoma cell line (SNB-19) human astrocytes	50 mg/kg for 2 days to chick embryos	Yes	B	[87,95]
NITD008	RNA-dependent RNA polymerase inhibitor	Vero cell line	50 mg/kg for 5 days i.p. to adult A129 mice	No	-	[96,97,98]
PHA-690509	cyclin-dependent kinase inhibitor	human neural progenitor cells induced-pluripotent neural stem cells	N.D.	No	-	[87]
Ribavirin	RNA-dependent RNA polymerase inhibitor	Vero cell line human neural progenitor cells	15 mg for 3 days i.p. to adult STAT-1 deficient mice	Yes	X	[88,91,99]
Sofosbuvir	RNA-dependent RNA polymerase inhibitor	human neuroepithelial stem cell placental cell line neuroblastoma cell line human fetal-derived neuronal stem cells induced-pluripotent neural stem cells brain organoids	33 mg/kg for 7 days p.o. to adult C57BL/6 mice 20 mg/kg for 7 days i.p. to neonatal Swiss mice	Yes	B	[16,100,101,102,103]
Z2	Direct inhibitor	Vero cell line BHK21 cell line	Single dose of 10 mg/kg i.p. to pregnant C57BL/6 mice Single dose of 10 mg/kg i.p. to A129 and AG6 adult mice	No	-	[104]

B.i.d.: bis in die (twice a day); ZIKV: Zika virus; s.c.: subcutaneous; i.m. intramuscular; i.p.: intraperitoneal; p.o.: per os; N.D. no data.

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
