# Peer review of "Late Neurological Consequences of Zika Virus Infection: Risk Factors and Pharmaceutical Approaches"

_pharmaceuticals, 2019, doi:10.3390/ph12020060_

Reviewer 1 Report

In this review, the authors described the epidemiology of ZIKV-associated neurological complications and the role of factors that may influence disease outcome. Moreover, the authors discuss experimental and clinical evidence of drugs that have shown promising results in vitro or in vivo against viral replication and and/or ZIKV-induced neurotoxicity. In overall, the manuscript is well written.

Major,

1, the authors list several drugs which can be potentially used for ZIKV. I suggest the authors can put the chemical structures of the drugs in the manuscript. In some cases, the complex structure may be also available. Probably the authors can also put this information in the manuscript.

Author Response

--As suggested by the reviewer, we have added the chemical structures of the drugs with potential anti-ZIKV activity as a new Supplementary Figure 1, in the revised version of our manuscript.

Reviewer 2 Report

The review paper by Souza and co-workers, entitled “Late neurological consequences of Zika virus 3 infection: risk factors and pharmaceutical approaches” is interesting and well written. Presented comprehensive information will be very useful for researchers working in the Zika field. Therefore I recommend this paper for publication in the current form.

Author Response

We thank the reviewer for his/her comments, and we are glad that he/she considered our manuscript “well written”.